# Comparative Performance Evaluation of an Accuracy-Enhancing Lyapunov Solver [†]

**Vasile Sima** 

National Institute for Research & Development in Informatics, 011455 Bucharest, Romania; vsima@ici.ro

† This paper is an extended version of our paper published in 2018 22nd International Conference on System Theory, Control and Computing (ICSTCC), Sinaia, Romania, 10–12 October 2018.

**Abstract:** Lyapunov equations are key mathematical objects in systems theory, analysis and design of control systems, and in many applications, including balanced realization algorithms, procedures for reduced order models, Newton methods for algebraic Riccati equations, or stabilization algorithms. A new iterative accuracy-enhancing solver for both standard and generalized continuous- and discrete-time Lyapunov equations is proposed and investigated in this paper. The underlying algorithm and some technical details are summarized. At each iteration, the computed solution of a reduced Lyapunov equation serves as a correction term to refine the current solution of the initial equation. The best available algorithms for solving Lyapunov equations with dense matrices, employing the real Schur(-triangular) form of the coefficient matrices, are used. The reduction to Schur(-triangular) form has to be done only once, before starting the iterative process. The algorithm converges in very few iterations. The results obtained by solving series of numerically difficult examples derived from the SLICOT benchmark collections for Lyapunov equations are compared to the solutions returned by the MATLAB and SLICOT solvers. The new solver can be more accurate than these state-of-the-art solvers and requires little additional computational effort.

**Keywords:** linear multivariable systems; Lyapunov equation; numerical algorithms; software; stability

## 1. Introduction

Lyapunov equations are key mathematical objects in systems theory, analysis and design of (control) systems, and in many applications. Solving these equations is an essential step in balanced realization algorithms [1,2], in procedures for reduced order models for systems or controllers [3–7], in Newton methods for algebraic Riccati equations (AREs) [8–14], or in stabilization algorithms [12,15,16]. Stability analyses for dynamical systems may also resort to Lyapunov equations.

Standard continuous-time or discrete-time Lyapunov equations,

$$A^T X + X A = -Y, \tag{1}$$

$$A^T X A - X = -Y, \tag{2}$$

respectively, with symmetric matrix $Y$, $Y = Y^T$, and $T$ denoting the matrix transposition, are associated to an autonomous linear time-invariant system, described by

$$\delta(x(t)) = Ax(t), \ t \geq 0, \ x(0) = x_0, \tag{3}$$

where $x(t) \in \mathbb{R}^n$, and $\delta(x(t))$ is either $dx(t)/dt$—the differential operator, or $x(t+1)$—the advance difference operator, respectively. A necessary and sufficient condition for asymptotic stability of system (3) is that for any symmetric positive definite matrix $Y$, denoted $Y > 0$, there is a unique

solution $X > 0$ of the Lyapunov Equation (1), or (2). Several other facts deserve to be mentioned. If $Y$ is positive-semidefinite, denoted $Y \geq 0$, and $X > 0$, then all trajectories of $x(t)$ in system (3) are bounded. If, in addition, the pair $(Y, A)$ is observable, then system (3) is globally asymptotically stable. Another sufficient condition for global asymptotic stability is that $Y > 0$ and $X > 0$. If $Y \geq 0$ and $X \not\geq 0$, then $A$ is not stable. If $V(x) = x^T X x$ is a *generalized energy*, it follows that $\frac{dV(x)}{dt} = -x^T Y x$, in the continuous-time case, and $V(x(t+1)) - V(x(t)) = -x^T Y x$, in the discrete-time case, that is, $x^T Y x$ is the associated *generalized dissipation*. The function $V(x)$ is a quadratic *Lyapunov function*. If $X > 0$, then $V(x) = 0$ implies $x = 0$.

For convenience, the often used notions and notation are given here.

- $\text{op}(M)$ is either $M$ or $M^T$, for a matrix $M$.
- $\mathcal{R}(M)$ is the residual matrix obtained when the unknown matrix $X$ is replaced by $M$ in an equation. For instance, for Equation (1), $\mathcal{R}(M) = A^T M + MA + Y$.
- Matrix $M$ is *(upper) quasi-triangular* if it is block (upper) triangular with diagonal blocks of order 1 or 2.
- Matrix $M$ is in a *real Schur form* if it is upper quasi-triangular and any $2 \times 2$ diagonal block has complex conjugate eigenvalues.
- Matrix $M \in \mathbb{R}^{n \times n}$ is in *Hessenberg form* if it has zeros under the first subdiagonal, i.e., $m_{ij} = 0, j = 1 : n := 1, 2, \ldots, n, i = j + 2 : n$.
- Matrix pair $(M, N)$ is in a *generalized real Schur form*, also named *real Schur-triangular form*, if $M$ is in a real Schur form and $N$ is upper triangular.
- Frobenius norm of a matrix $M \in \mathbb{R}^{n \times p}$ is $\|M\|_F := \sqrt{\sum_{j=1}^{p} \sum_{i=1}^{n} m_{ij}^2} = \sqrt{\sum_{i=1}^{\min\{n,p\}} \sigma_i^2}$, where $\sigma_i$, $i = 1 : \min\{n, p\}$ are the singular values of $M$. If $M = M^T$, $\|M\|_F = \sqrt{\sum_{i=1}^{n} \lambda_i^2}$, where $\lambda_i \in \Lambda(M)$, $i = 1 : n$, are the eigenvalues of $M$.
- `rcond`: estimated reciprocal condition number.
- $\varepsilon_M$: relative machine precision, $\varepsilon_M \approx 2.22 \times 10^{-16}$, in double precision format (IEEE 754 standard).

This paper considers *generalized* continuous-time or discrete-time Lyapunov equations

$$\text{op}(A)^T X \,\text{op}(E) + \text{op}(E)^T X \,\text{op}(A) \quad = \quad -Y, \tag{4}$$

$$\text{op}(A)^T X \,\text{op}(A) - \text{op}(E)^T X \,\text{op}(E) \quad = \quad -Y, \tag{5}$$

respectively, where $A, E \in \mathbb{R}^{n \times n}$. The operator $\text{op}(M)$ is often used in basic numerical linear algebra software [17,18], for increased generality and flexibility. A necessary solvability condition is that both $A$ and $E$, for Equation (4), or either $A$ or $E$, for Equation (5), are nonsingular. It will be assumed, without loss of generality, that $E$ in Equation (5) is nonsingular. If $\Lambda(A, E) \subset \mathbb{C}_-$, where $\Lambda(M, N)$ is the set of eigenvalues of the matrix pencil $M - \lambda N$, $\lambda \in \mathbb{C}$, and $\mathbb{C}_-$ is the open left half of the complex plane $\mathbb{C}$, in the continuous-time case, or the open unit disk centered in the origin of $\mathbb{C}$, in the discrete-time case, then Equations (4) or (5) are *stable* Lyapunov equations. If $Y \geq 0$, a stable Lyapunov equation has a unique solution $X \geq 0$, that can be expressed and computed in a factored form, $X = U^T U$, where $U$ is the Cholesky factor of $X$ [19]. The *standard* Lyapunov Equations (1) or (2) are special cases of the generalized Equations (4) or (5), where $E$ is an identity matrix, $E = I_n$, and $\text{op}(M) = M$.

There are applications for which the availability of the $\text{op}(\cdot)$ operator is important. Such an application is the computation of the Hankel singular values of a dynamical system,

$$E\delta(x(t)) = Ax(t) + Bu(t), \quad y(t) = Cx(t), \tag{6}$$

for which, two related Lyapunov equations are defined,

$$APE^T + EPA^T = -BB^T, \quad A^T QE + E^T QA = -C^T C, \tag{7}$$

in the continuous-time case, and

$$APA^T - EPE^T = -BB^T, \quad A^TQA - E^TQE = -C^TC, \tag{8}$$

in the discrete-time case.

The solutions $P$ and $Q$ of these equations are the *controllability* and *observability Gramians*, respectively, of system (6). The *Hankel singular values* are the nonnegative square roots of the eigenvalues of the matrix product $QP$. If the system (6) is stable, then $P \geq 0$ and $Q \geq 0$, and these properties imply that $QP \geq 0$. But these theoretical results may not hold in numerical computations if the symmetry and semidefiniteness are not preserved by the solver. Some computed Hankel singular values may be returned as negative or even complex numbers. Such an example is given in [20]. This proves how important is to ensure the accuracy and reliability of the results. The recommended algorithm for this application, proposed in [19], for $E = I_n$, and extended in [21] for a general matrix $E$, uses $B$ and $C$ directly, without evaluating $BB^T$ and $C^TC$, and computes the Choleky factors $R_c$ and $R_o$ of the Gramians, $P = R_cR_c^T$, $Q = R_o^TR_o$. The Hankel singular values are then obtained as the singular values of the product $R_cR_o$, which are all real nonnegative.

Many algorithms have been proposed to solve Lyapunov and more general linear matrix equations. The first numerically stable algorithm has been developed by Bartels and Stewart in [22] for *Sylvester equation*, $AX + XB = C$, where $A \in \mathbb{R}^{n \times n}$, $B \in \mathbb{R}^{m \times m}$, and $C \in \mathbb{R}^{n \times m}$, and also specialized for solving Lyapunov Equation (1). A *transformation approach* is used: $A^T$ and $B$ are each reduced to a *quasi-triangular form*, using orthogonal transformations $U$ and $V$, $\widetilde{A} = U^TA^TU$, $\widetilde{B} = V^TBV$, and $C$ is updated, $\widetilde{C} = U^TCV$. Then, a reduced equation, $\widetilde{A}^T\widetilde{X} + \widetilde{X}\widetilde{B} = \widetilde{C}$ is solved by a special back substitution process, and its solution is transformed back to the solution of the original equation, $X = U\widetilde{X}V^T$. For standard Lyapunov equations, $A$ is reduced to a quasi-triangular form or a *real Schur form*, but the rest of the procedure is similar. A more efficient algorithm for Sylvester equations with $n \geq m$ is based on the Hessenberg-Schur method [23], which reduces $B^T$ to quasi-triangular form and $A$ to *Hessenberg form*. Clearly, this algorithm has no advantage for Lyapunov equations. Hammarling's algorithm [19] also uses the transformation approach for stable Lyapunov equations with $Y \geq 0$, and computes the Cholesky factor of the solution. Many algorithmic and computational details for Sylvester and standard Lyapunov equations are given, e.g., in [12]. Computational improvements for solving the reduced equations have been proposed in [24–26]. An extension of Bartels-Stewart algorithm for generalized Lyapunov equations has been described in [21]. In this case, using two orthogonal matrices, the pair $(A, E)$ is reduced to the *generalized real Schur form* [27], also called *real Schur-triangular form*, $(\widetilde{A}, \widetilde{E})$, with $\widetilde{A}$ in a real Schur form and $\widetilde{E}$ upper triangular. Then, the right hand side $Y$ is updated accordingly, the corresponding reduced Lyapunov equation is solved, and the result is transformed back to the solution of the original equation. A comprehensive recent survey of the theory and applications of linear matrix equations is [28].

It is worth to mention that solvers implementing Bartels-Stewart-like approaches can be used for small and medium size Lyapunov equations, with $n$ currently less than a few thousands, due to their complexity of order $n^3$. Large-order equations can be approached by iterative algorithms, usually exploiting sparsity and/or the low-rank structure, see [28] and the references therein. A compact conjugate-gradient algorithm is proposed in [29] for solving large-scale Equation (4) with factored, low-rank $Y$ and symmetric positive definite matrices $A$ and $E$. Iterative methods recorded a fast development in recent years for solving various linear and nonlinear problems. For instance, Kyncheva et al. [30] analyze the local convergence of Newton, Halley and Chebyshev iterative methods for simultaneous determination of all multiple zeros of a polynomial function over an arbitrary normed field, while [31] presents a new semi-local convergence analysis for Newton's method in a Banach space for systems of nonlinear equations.

This paper investigates the accuracy and efficiency of several Lyapunov solvers for equations with dense matrices. Specifically, the state-of-the-art solvers from the Control System Toolbox [32] and SLICOT Library [20,33,34] (www.slicot.org), and a new accuracy-enhancing iterative solver, referred to

as ArLyap, are considered. As in [35], the ArLyap solver has been derived as a special case of an ARE solver based on Newton's method, with or without line search [13,14,36–38]. Actually, Lyapunov equations are simplified AREs, without the quadratic or rational matrix term. All these solvers are based on the best algorithms for Lyapunov equations with dense matrices: the algorithm in [22] and its generalization [21], both available in SLICOT. Relatively straitforward modifications of the ArLyap solver allow to use other algorithms for solving the reduced equations, for instance, Hammarling's or Penzl's algorithms in [19] or [21], respectively, for stable equations with $Y \geq 0$.

The ArLyap solver offers an option for specifying an initial approximation, $X_0$. It is possible, for instance, to use some upper or lower bounds of the solution, derived as described in [39]. Using tighter estimates may reduce the number of iterations for convergence. Another option is to use the op operator, enherited from the lower-level SLICOT solvers. This allows to compute the real Schur-triangular form of the pair $(A, E)$ (or the real Schur form of $A$, when $E = I_n$) only once for obtaining both controllability and observability Gramians. It is not necessary to do these computations for $(A^T, E^T)$ (or $A^T$).

This paper extends the developments in [35] by using a specialized, more efficient algorithm, which iterates directly on reduced Lyapunov equations, with $A$ in a real Schur form, and $E$ upper triangular. The main computational modules involved, which are not available in BLAS [17] or LAPACK [18] Libraries, are also discussed.

The paper is structured as follows. Section 2 presents the numerical results for solving series of test examples from the SLICOT benchmark collections for Lyapunov equations, CTLEX [40] and DTLEX [41]. Section 3 further discusses the relevance of these results. Section 4 describes the underlying algorithm and the new computational modules. Section 5 concludes the paper.

## 2. Results

This section presents several results illustrating the performance of the accuracy-enhancing Lyapunov solver, ArLyap, in comparison to the state-of-the art Control System Toolbox [32] and SLICOT Library solvers. ArLyap solves reduced Lyapunov equations at each iteration. The same computational environment as in [35] has been used (64-bit Intel Core i7-3820QM, 2.7 GHz, 16 GB RAM, double precision, Intel Visual Fortran Composer XE 2015 and MATLAB 8.6.0.267246 (R2015b), Natick, MA, USA). An executable MATLAB® MEX-function has been linked using ten new subroutines, several SLICOT subroutines, and optimized LAPACK and BLAS libraries included in MATLAB. The results presented in this section and the next one are new, and complement those reported in [35].

### 2.1. Benchmark Examples

To make possible a comparison with previous results, obtained with the ALyap solver and reported in [35], the same SLICOT benchmark collections for Lyapunov equations, CTLEX [40] and DTLEX [41], have been used. These benchmarks allow to investigate the behavior of numerical methods in difficult situations and assess their correctness, accuracy, and speed. The collections contain parameter-dependent examples of scalable size (group 4). For convenience, the short notation TLEX will be used for both collections and their examples. TLEX 4.1 and TLEX 4.2 define stable standard Lyapunov equations, while TLEX 4.3 and TLEX 4.4 define generalized Lyapunov equations (stable for TLEX 4.4). Moreover, the solutions of the equations for TLEX 4.1 and TLEX 4.3 are considered known (being computable with machine accuracy).

TLEX examples are generated using several parameters: the order $n$, and parameters $r$, $s$, $\lambda$, and $t$, which define the *numerical condition* of the problem, that influences the accuracy of the solution and its sensitivity to small perturbations in the data matrices. Increasing the value of any of these parameters, including $n$, makes the problem more ill-conditioned. Very ill-conditioned examples can be built even for small values of $n$. The same values of these parameters as in [35] have been used (see Table 2 in [35]). Specifically, the sets of values for $n$, $r$, $s$, $\lambda$, and $t$ are defined by the following lists:

list_n $= 5:5:20$       for TLEX 4.1 – TLEX 4.3;     list_n $= 15:15:60$ for TLEX 4.4;

list_r $= 1.1:0.2:1.9$     for TLEX 4.1;

list_s $= 1.1:0.2:1.9$     for TLEX 4.1 and TLEX 4.2;

list_l $= -2:0.2:-0.2$   for CTLEX 4.2          list_l $= -0.9:0.2:0.9$ for DTLEX 4.2;

list_t $= 1:1:30$         for TLEX 4.3            list_t $= 1.1:0.2:9.9$ for TLEX 4.4;

where the notation in MATLAB style $i = k:l:m$ means that $i$ takes the values $k, k+l, k+2l, \ldots, m$.

A series of equations has been generated for each TLEX example, using two or three nested loops. The series for TLEX 4.1 is produced by a loop for n = list_n, incorporating a loop for r = list_r, containing, in turn, a loop for s = list_s. The order of the loops is list_n, list_l, and list_s, for TLEX 4.2, and list_n and list_t, for TLEX 4.3 and TLEX 4.4. Each abscissa value in the figures below is the index of an example in a generated series. All figures in this paper are new.

## 2.2. Performance Analysis Issues

The accuracy of a computed solution, $\hat{X}$, is assessed using the *relative error*,

$$e(\hat{X}) := \|\hat{X} - X\|_F / \max(1, \|X\|_F),$$

when the true solution, $X$, is known (i.e., for TLEX 4.1 and TLEX 4.3). In this formula, $\|M\|_F$ denotes the Frobenius norm of the matrix $M$. If $X$ is unknown (i.e., for TLEX 4.2 and TLEX 4.4), the *normalized residual* with respect to $X_m$, defined as

$$r(\hat{X}) := \|\mathcal{R}(\hat{X})\|_F / \max(1, \|X_m\|_F), \tag{9}$$

is used by the performance analysis program, where $\mathcal{R}(\hat{X})$ is the residual matrix at $\hat{X}$ (see the definition of $\mathcal{R}(\cdot)$ in Section 1), and $X_m$ is the solution computed by the MATLAB functions lyap or dlyap, for CTLEX and DTLEX examples, respectively. The usual definition of the normalized residual, used internally by the ArLyap solver to decide if convergence has been achieved, has $\hat{X}$ instead of $X_m$ in its denominator. The use of $X_m$ in Equation (9) allows to make fair comparisons of the residual norms corresponding to all these solvers.

In order to avoid too ill-conditioned examples, which cannot be reliably solved by any solver, the performance analysis program also estimates the reciprocal condition number, rcond, of Lyapunov equations, and may bound its value. The SLICOT-based MATLAB functions lyapcond and steicond are used as condition estimators for standard continuous- and discrete-time Lyapunov equations, respectively. The same functions are called for generalized Lyapunov equations, by replacing the matrices $A$ and $Y$ by $E^{-1}A$ and $E^{-T}YE^{-1}$, respectively. These estimators are using the exact solution $X$, when known, or the MATLAB computed solution $X_m$, otherwise. The chosen sequence of parameter values for each example produces a zigzaggy variation of rcond.

As in [35], the equations with an estimated rcond smaller than $\varepsilon_M^{1/2} \approx 1.49 \times 10^{-8}$, where $\varepsilon_M$ is the *relative machine precision*, have been excluded for TLEX 4.1 and CTLEX 4.4 series of examples. Similarly, the equations with rcond $< 10^{-14}$ have been omitted from the TLEX 4.2 generated series. Moreover, the examples with $\|X_m\|_F > 0.4/\varepsilon_M \approx 1.8 \times 10^{15}$ have been excluded from the DTLEX 4.4 series.

## 2.3. Continuous-Time Lyapunov Equations

Figure 1 shows the relative errors for CTLEX 4.1 series of examples generated as mentioned in Section 2.1. The MATLAB and SLICOT solvers are compared with the ArLyap solver. Figure 2 depicts the number of iterations for ArLyap, the estimated reciprocal condition numbers, rcond, of the equations, and the Frobenius norms of the true solutions. Almost all examples needed less than four iterations. Figure 3 plots the elapsed CPU times of the three solvers for this series of examples. The SLICOT solver is the fastest, and it is closely followed by the ArLyap solver. The accuracy results (Figure 1) for well-conditioned equations, i.e., with rcond close to 1, are slightly worse than those

reported in [35], for reasons explained in detail in Section 3. However, for several ill-conditioned examples, such as those numbered 50, 54, 55, or 63, the relative errors for ArLyap are much smaller than for the ALyap solver.

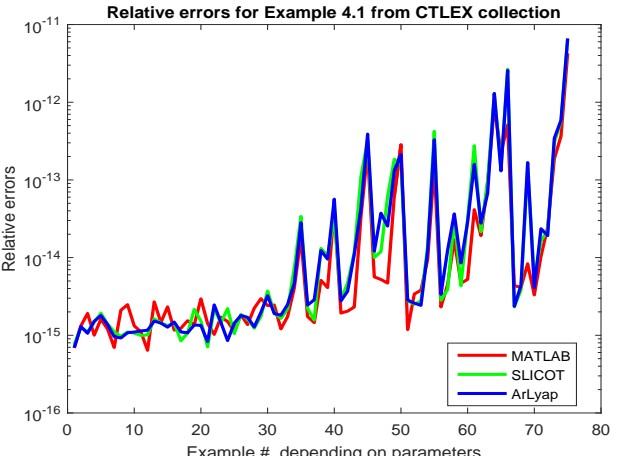

**Figure 1.** Relative errors for CTLEX 4.1 series of examples.

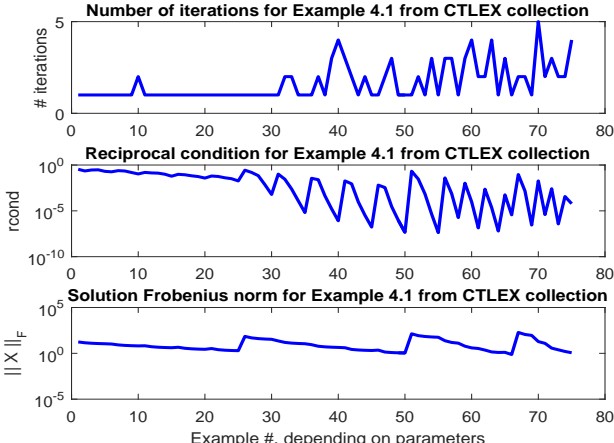

**Figure 2.** Number of iterations for ArLyap solver, reciprocal condition numbers, and the known solution norms for CTLEX 4.1 series of examples.

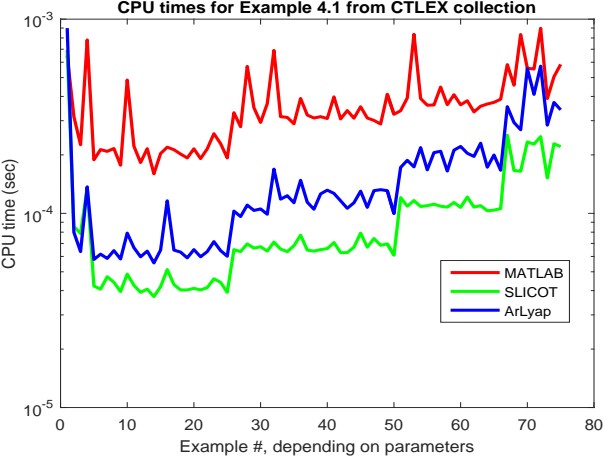

**Figure 3.** Elapsed CPU time for CTLEX 4.1 series of examples.

Table 1 shows the values, rounded to three significant digits, of the normalized residuals for three examples of the CTLEX 4.1 series, using ALyap and ArLyap. The ArLyap solver obtained smaller normalized residuals, possibly in fewer iterations, than the ALyap solver. But the difference is that the pairs of the sets of values in Table 1 are for the solutions of original Lyapunov equations, and for reduced Lyapunov equations, respectively.

**Table 1.** Normalized residuals during iteration of ALyap and ArLyap solvers for three examples of the CTLEX 4.1 series.

| Parameters | Algorithm | Normalized Residuals |
|---|---|---|
| $n = 5, r = s = 1.1$ | ALyap | $47.1, 1.49 \times 10^{-15}, 2.39 \times 10^{-16}$ |
| | ArLyap | $47.1, 2.20 \times 10^{-16}, 1.58 \times 10^{-16}$ |
| $n = 10, r = s = 1.3$ | ALyap | $188, 9.86 \times 10^{-15}, 2.47 \times 10^{-15}, 1.50 \times 10^{-15}$ |
| | ArLyap | $188, 1.26 \times 10^{-15}, 8.54 \times 10^{-16}$ |
| $n = 20, r = 1.5, s = 1.3$ | ALyap | $852, 1.37 \times 10^{-11}, 9.36 \times 10^{-13}, 6.06 \times 10^{-13}, 5.77 \times 10^{-13}$ |
| | ArLyap | $852, 3.79 \times 10^{-14}, 3.65 \times 10^{-14}$ |

As shown in Figure 1, the MATLAB function `lyap` sometimes obtained smaller normalized residuals than the other solvers for the CTLEX 4.1 series of examples. But `lyap` and `dlyap` use a balancing procedure before computing the real Schur(-triangular) form. The current results for SLICOT and ArLyap solvers are computed without any balancing. Moreover, `lyap` is in advantage in a comparison since all computations are done on the given data, while the other solvers get the matrices from the MATLAB context. But even this transfer involves a loss of accuracy. For instance, for CTLEX 4.1 with $n = 3$, and $r = s = 1.9$, the relative error between the MATLAB and Fortran representations of the matrix $A$ is about $1.47 \times 10^{-15}$, that is, almost one digit of accuracy has been lost. The matrix $Y$ lost less accuracy, since its relative error is about $8.52 \times 10^{-16}$. Therefore, the data matrices used by the solvers are not exactly the same.

The normalized residuals for CTLEX 4.2 series of examples are slightly worse than in [35]. Figures 4 and 5 plot other performance results, as for CTLEX 4.1 series. Very few iterations are made. The `rcond` values can be even smaller than $10^{-10}$, and the solution can have a large Frobenius norm. The ArLyap solver is faster than the MATLAB function `lyap`, with few exceptions, and slightly slower than the SLICOT solver.

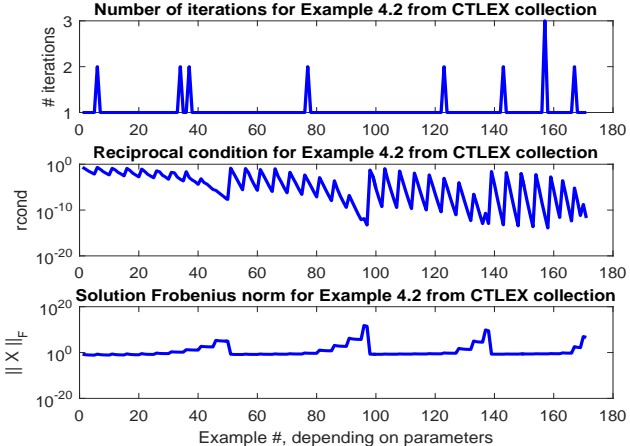

**Figure 4.** Number of iterations for ArLyap solver, reciprocal condition numbers, and computed solution norms for CTLEX 4.2 series of examples.

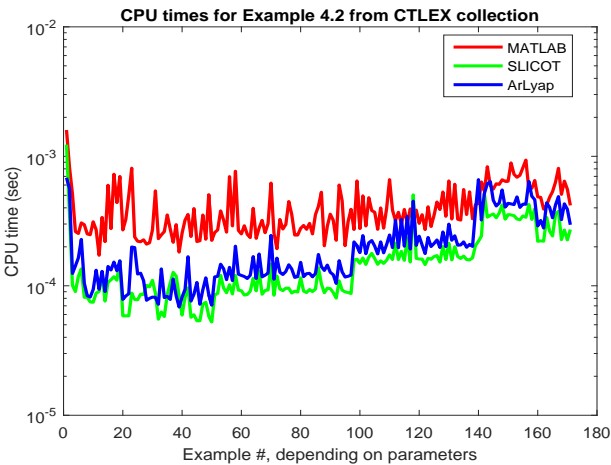

**Figure 5.** Elapsed CPU time for CTLEX 4.2 series of examples.

Figure 6 plots the relative errors for CTLEX 4.3 series of examples. The ArLyap solver has almost identical errors as the SLICOT solver, clearly (much) smaller than lyap for most examples.

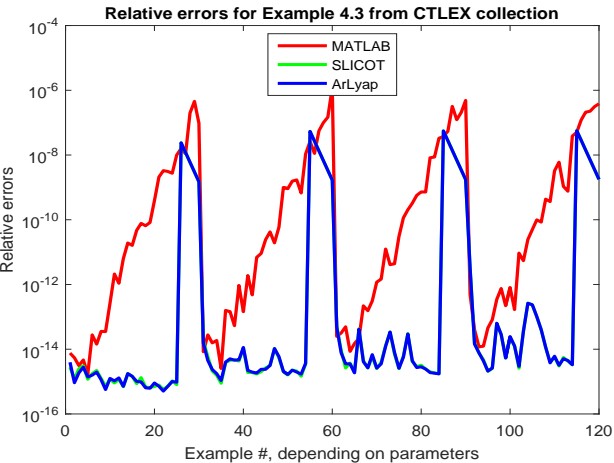

**Figure 6.** Relative errors for CTLEX 4.3 series of examples.

The normalized residuals for the three solvers have comparable values for CTLEX 4.4 series, see Figure 7. For most examples, one iteration has been taken, as shown in Figure 8. The normalized residuals are clearly influenced by the condition numbers.

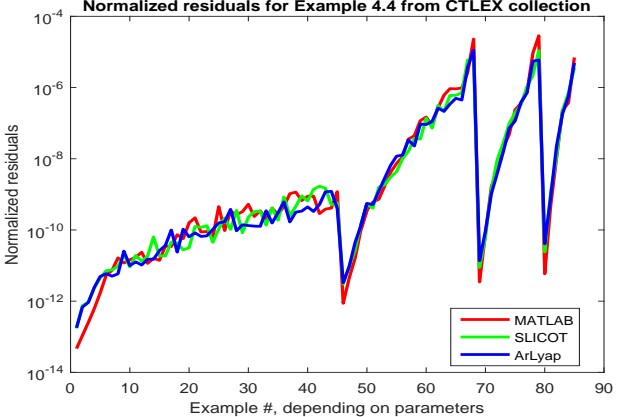

**Figure 7.** Normalized residuals for CTLEX 4.4 series of examples.

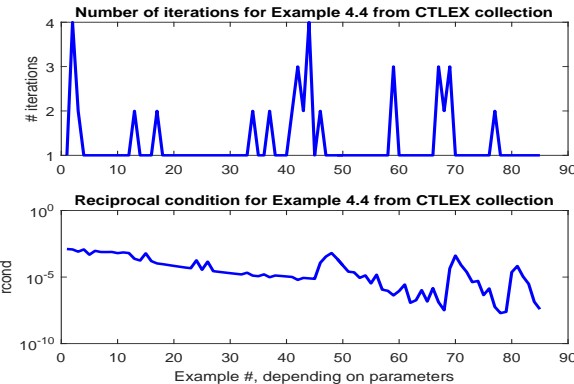

**Figure 8.** Number of iterations for ArLyap solver and reciprocal condition numbers for CTLEX 4.4 series of examples.

## 2.4. Discrete-Time Lyapunov Equations

Figures 9 and 10 show the relative errors and the number of iterations plus reciprocal condition numbers, respectively, for the DTLEX 4.1 series. A smaller internal tolerance, $\varepsilon_M^2$, has been used for deciding the convergence of the iterative process. This allowed to make additional iterations than usual in several cases, and reduce the errors. As for the CTLEX 4.1 series (see Figure 1), ArLyap and SLICOT solvers have comparable relative errors for well-conditioned equations, and hence worse than those reported in [35], but for several ill-conditioned examples, such as those numbered 55, 62, 65, 70–73, the relative errors for ArLyap are (much) smaller than those for the SLICOT solver and sometimes also than for ALyap. Figure 11 plots the elapsed CPU times for the three solvers.

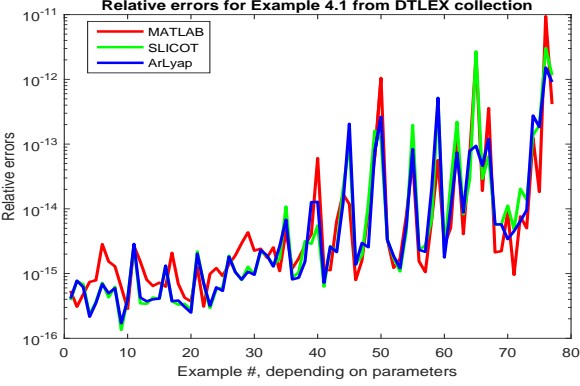

**Figure 9.** Relative errors for DTLEX 4.1 series of examples; tolerance $\varepsilon_M^2$.

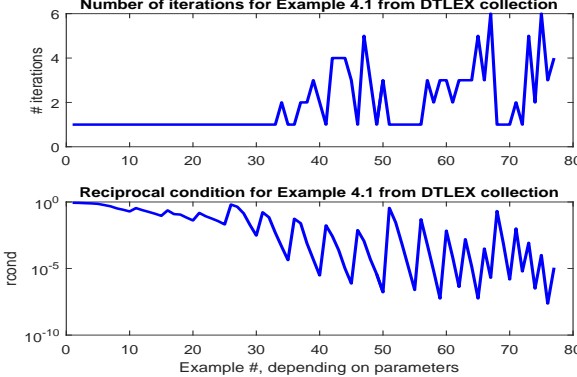

**Figure 10.** Number of iterations for ArLyap solver and reciprocal condition numbers for DTLEX 4.1 series of examples; tolerance $\varepsilon_M^2$.

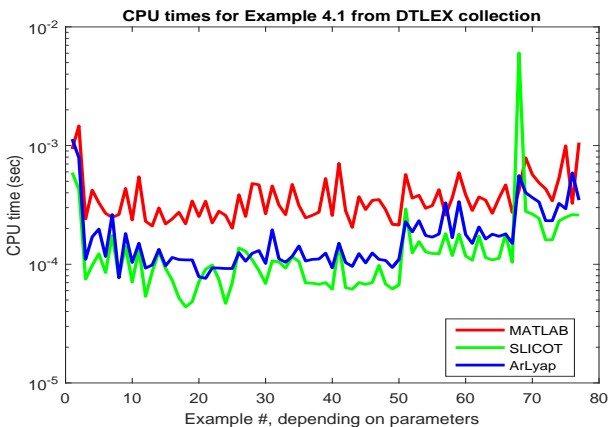

**Figure 11.** Elapsed CPU time for DTLEX 4.1 series of examples.

Figures 12 and 13 illustrate the performance of the solvers for the DTLEX 4.2 series. The ArLyap solver returned after the first iteration in most cases. Consequently, its accuracy is comparable to that of the SLICOT solver. Again, ArLyap is generally more accurate than ALyap for ill-conditioned equations, but not for well-conditioned ones.

Figure 14 shows the relative errors for the DTLEX 4.3 series of examples. The SLICOT and ArLyap solvers have comparable errors, which are often better, and sometimes much better, than the errors of the MATLAB function `dlyap`. However, the ALyap solver was more accurate than ArLyap for many examples.

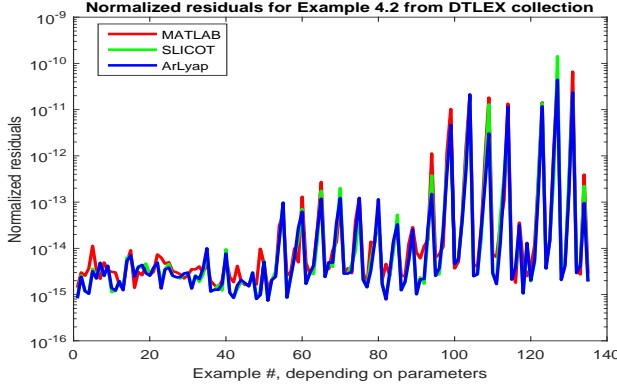

**Figure 12.** Normalized residuals for DTLEX 4.2 series of examples.

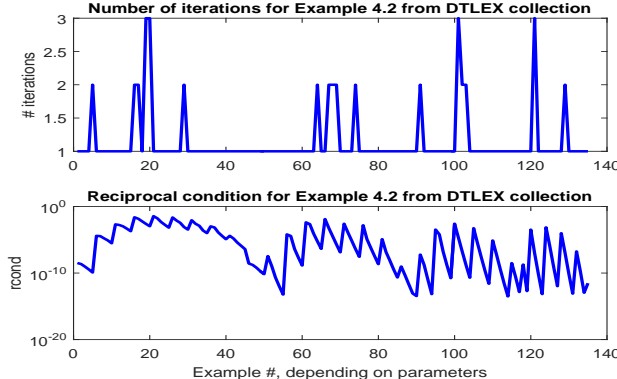

**Figure 13.** Number of iterations for ArLyap solver and reciprocal condition numbers for DTLEX 4.2 series of examples.

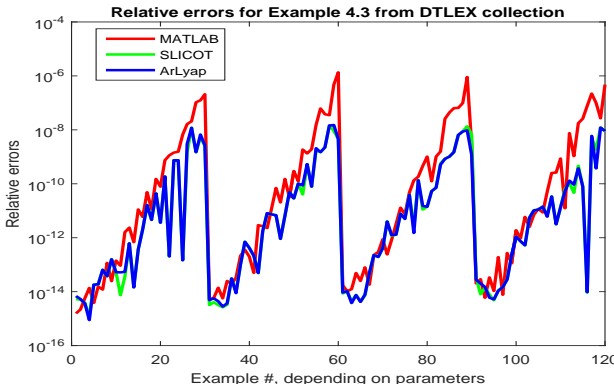

**Figure 14.** Relative errors for DTLEX 4.3 series of examples.

Almost always, the ArLyap solver obtained much smaller relative residuals for the DTLEX 4.4 series of examples, as shown in Figure 15. In this case, the matrix $X_0$ has been chosen as $X_m$, the solution computed by `dlyap`, in order to test the behavior for an initialization different from a zero matrix. In addition, the tolerance $\tau$ has been set to $\varepsilon_M^2$. This resulted in a larger number of iterations than usual for several examples, see Figure 16. The maximum number of iterations has been set to $k_{\max} = 10$. It should be mentioned that $\|X_m\|_F$ is very big for the examples needing ten iterations. For instance, $\|X_m\|_F \approx 1.58 \times 10^{15}$ for the last example in the series. If $\|X_m\|_F$ is limited to about $10^{-3}/\varepsilon_M \approx 4.5 \times 10^{12}$, the maximum number of iterations is seven (for two examples only) and the maximum normalized residual is $9.7 \times 10^{-13}$.

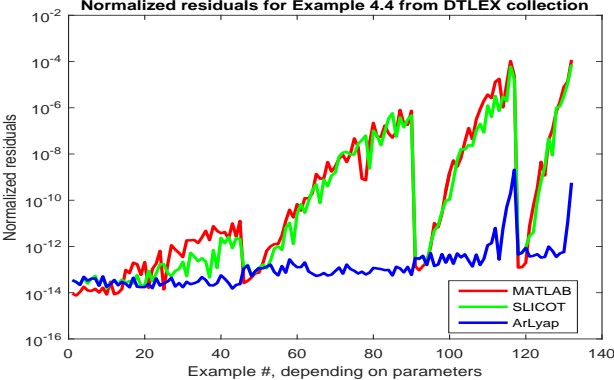

**Figure 15.** Normalized residuals for DTLEX 4.4 series of examples; tolerance $\varepsilon_M^2$. The ArLyap solver is initialized by `dlyap` solution.

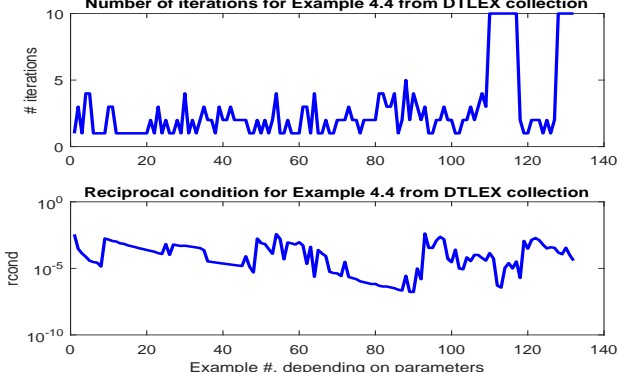

**Figure 16.** Number of iterations for ArLyap solver, initialized by `dlyap` solution, and reciprocal condition numbers for DTLEX 4.4 series of examples; tolerance $\varepsilon_M^2$.

## 3. Discussion

The ArLyap solver differs from its previous version, ALyap, dealt with in [35], by solving reduced Lyapunov equations at each iteration, without back transforming their solutions. This implied the use of the real Schur form of the matrix $A$, or of the real Schur-triangular form of the matrix pair $(A, E)$ for residual matrix computation, which provided gains in efficiency, and expected gains in accuracy, by exploiting the (almost) triangular structure of these matrices. More details will be given in Section 4. However, the numerical results have shown slightly worse accuracy for some equations in TLEX 4.1–TLEX 4.3 series of examples. To discover the reason for this abnormal behavior, the ArLyap solver has been modified to return the normalized residual computed for the original Lyapunov equation corresponding to the last iteration, in addition to the normalized residual for the reduced Lyapunov equation. The trajectories for both types of normalized residuals are plotted in Figures 17–21, for CTLEX 4.1, CTLEX 4.2, CTLEX 4.4, DTLEX 4.2, and DTLEX 4.3 series, respectively. The allowed reciprocal condition numbers have not been restricted for CTLEX 4.1 examples, and therefore more (also more ill-conditioned) equations have been solved. A similar behavior has been seen for the other TLEX series of examples. Clearly, there is a significant difference, usually of one, and sometimes even more, orders of magnitude between the corresponding points of the two trajectories for each series.

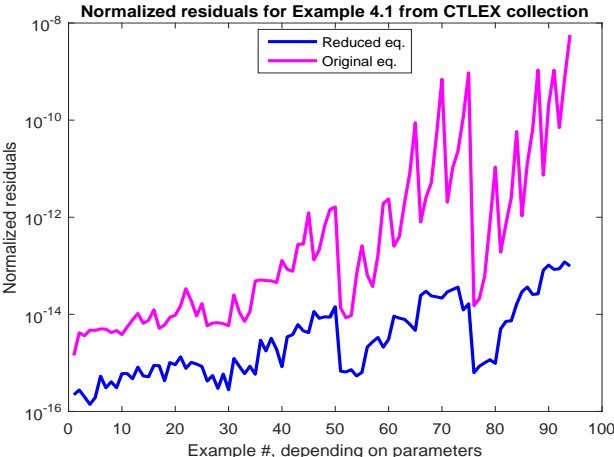

**Figure 17.** Normalized residuals for reduced and original Lyapunov equations for CTLEX 4.1 series of examples, using the ArLyap solver.

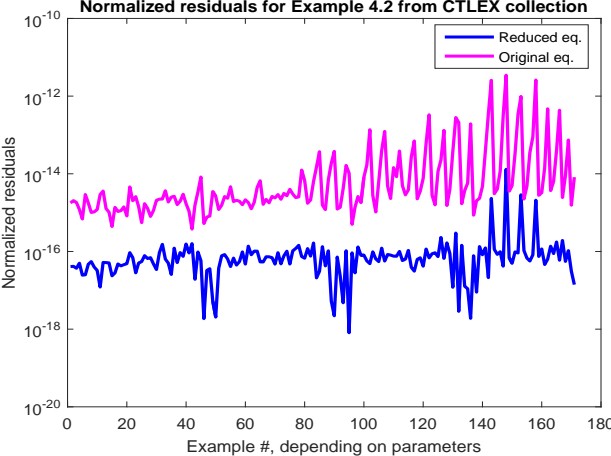

**Figure 18.** Normalized residuals for reduced and original Lyapunov equations for CTLEX 4.2 series of examples, using the ArLyap solver; tolerance $\varepsilon_M^2$.

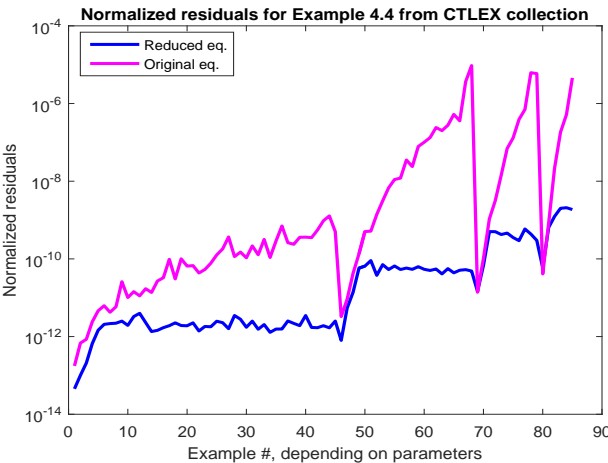

**Figure 19.** Normalized residuals for reduced and original Lyapunov equations for CTLEX 4.4 series of examples, using the ArLyap solver.

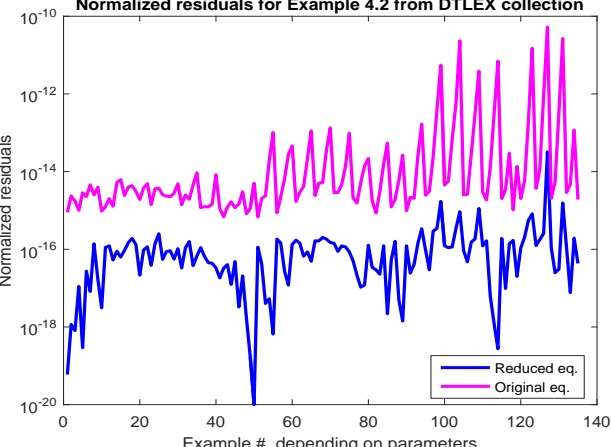

**Figure 20.** Normalized residuals for reduced and original Lyapunov equations for DTLEX 4.2 series of examples, using the ArLyap solver; tolerance $\varepsilon_M^2$.

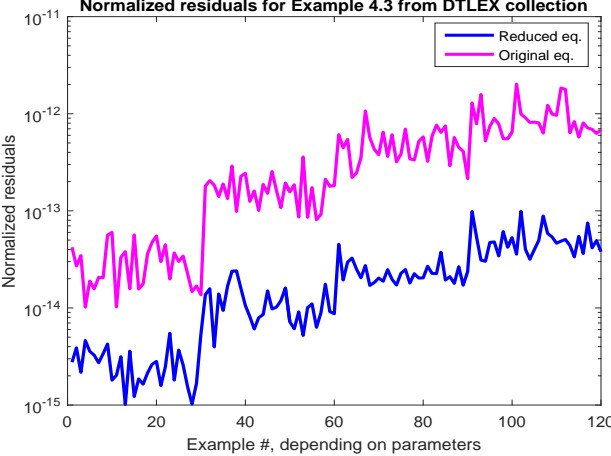

**Figure 21.** Normalized residuals for reduced and original Lyapunov equations for DTLEX 4.3 series of examples, using the ArLyap solver.

Currently, the ArLyap solver uses the normalized residuals for the reduced Lyapunov equations to decide convergence. While these residuals should theoretically coincide to those for the original

equations, there is a big discrepancy between their numerical values. In addition, less iterations are needed for deciding that the convergence has been achieved. These issues could make the final errors or residuals (computed by the external MATLAB program, not by the solver) to be sometimes larger than those obtained using the ALyap solver. It can be seen that in most cases the trajectories of the normalized residuals for the original equations are comparable in shape and magnitude to the trajectories of the relative errors or normalized residuals computed externally, and shown in the previous section. It should be emphasized that this increase in the normalized residual values is produced just by the back transformation (with orthogonal matrices!) of the solutions of reduced Lyapunov equations obtained at the end of the iterative process, and by recomputing the residuals using $A$ (or $A$ and $E$) in Equations (1) or (2) (or in (4) or (5)).

Even in computations with orthogonal matrices, rounding errors can significantly perturb the results. For instance, using the first CTLEX 4.1 example in the generated series, with $n = 5$, $r = s = 1.1$, if $Q$ is the orthogonal matrix reducing $A$ to a real Schur form, $\widetilde{A} = Q^T A Q$, and $\widetilde{Y} = Q^T Y Q$ is the transformed matrix $Y$ in Equation (1), then $\|Q\widetilde{Y}Q^T - Y\|_F \approx 4.32 \times 10^{-14}$, and $\|Q\widetilde{Y}Q^T - Y\|_F / \|Y\|_F \approx 9.17 \times 10^{-16}$, while these values should theoretically be zero. If $\widetilde{X}$ is the solution of the corresponding reduced Lyapunov equation, $\widetilde{A}^T \widetilde{X} + \widetilde{X}\widetilde{A} = -\widetilde{Y}$, computed using MATLAB function `lyap`, its normalized residual is about $3.39 \times 10^{-16}$, but the normalized residual of the solution of the original equation, $X = Q\widetilde{X}Q^T$ is about $1.67 \times 10^{-15}$, i.e., about five times bigger than for $\widetilde{X}$. This increase is produced by the two multiplications, with $Q$ and $Q^T$. Similarly, for the last CTLEX 4.1 example in the generated series, with $n = 20$, $r = 1.9$, $s = 1.1$, the normalized residual for $X$ is over 307 times bigger than for $\widetilde{X}$. Such residual magnification could be attenuated only by using computations with extended precision.

To prove that the back transformation step increases the normalized residuals, the CTLEX 4.1 series of examples has been solved by ArLyap with the additional condition to exit after the first iteration. The ratios between the corresponding normalized residuals for the original and reduced equations have been computed. While the normalized residuals in these two cases should theoretically coincide, the computed values had ratios in the interval $[4.62, 420.43]$, with a mean value of about 54.78. This proves that the back transformation step always increased the normalized residuals, possibly by more than two orders of magnitude. However, the relative errors of the two solvers for this test are comparable. Specifically, the ratios of these errors for the ArLyap and SLICOT solvers are in the interval $[0.203, 2.67]$, with a mean value of about 1.04. There are 38 examples for which ArLyap was more accurate, and 37 examples for which it was less accurate.

As shown before, even if the normalized residual of the last iterate computed by ArLyap, $\widetilde{X}_k$, is very small, the normalized residual of the computed solution of the original equation, $Q\widetilde{X}_k Q^T$, can be much larger. The previous version of the accuracy-enhancing solver, ALyap, could sometimes achieve more accurate results, with some additional computational effort, by iterating directly on the matrices $Q\widetilde{X}_j Q^T$, $j = 0 : k$. Usually, the residual matrices of its iterates, and hence the corrections applied in the process, have larger norms than for the ArLyap solver (see Table 1).

For most of the tests, the default tolerance has been used, to make comparisons with [35] possible. However, the ArLyap solver produces smaller normalized residuals during iterations than the ALyap solver. Consequently, ArLyap can often return after one or two iterations. Indeed, with the default tolerance, all 75 examples generated for the CTLEX 4.1 series needed a total of 124 iterations, hence the mean value is about 1.65 iterations. This suggested to use a smaller tolerance, hoping for more accurate final results. With a tolerance $\tau = 10^{-6}\varepsilon$, 165 iterations were required, i.e., the mean value increased to 2.17. For both tolerance values above, the maximum number of iterations was five. Some results have been slightly improved, but not the global statistics, such as the mean of normalized residuals for the series of examples. Exactly the same results have been obtained with $\tau = 10^{-14}\varepsilon$. The reason is that there is an internal test preventing further iterations if the normalized residual increased from one iteration to the next one. In such a case, the previous iterate is restored and returned as the solution. Such an increase is often a sign that the limiting accuracy has been attained, and further iterations

could be purposeless. Even if the residual could be further decreased by chance, such a decrease will be rather small and will not justify spending additional computational effort. Further numerical experiments confirmed this conclusion. Indeed, the calculations have been repeated using a test which enabled the iterative process to continue if the current normalized residual is smaller than ten times the previous normalized residual value. But then, the normalized residual trajectory may either arrive to a constant value, or behave periodically, or have all further values in a small range. An exception occurred for the CTLEX 4.2 example, with $n = 10, \lambda = -0.6, s = 1.5$. The normalized residual had the following values during iterations

$$6.72 \cdot 10^{-1}, 1.04 \cdot 10^{-16}, 1.90 \cdot 10^{-16}, 9.72 \cdot 10^{-17}, 2.45 \cdot 10^{-17}, 1.91 \cdot 10^{-16},$$
$$1.92 \cdot 10^{-17}, 1.91 \cdot 10^{-16}, 6.73 \cdot 10^{-18}, 9.07 \cdot 10^{-18}, 5.67 \cdot 10^{-18},$$

showing that it increased three times. After each increase, the values decreased in the next one or two iterations. The last value is the smallest. The typical situation is, however, that the normalized residual at the iteration before the first such increase is either the minimum, or at most four times larger than the minimum, but often it is much closer.

It is almost impossible to find the best strategy for deciding when to stop. Sometimes, after a local increase of the normalized residual, the next few iterations will continuously decrease its value, but then another increase could appear, and the previously found minimum value could not be further reduced. Since the normalized residuals trajectory is optionally returned by the ArLyap solver, one possible strategy would be to find the minimum normalized residual value, and call the solver again with the maximum number of iterations, $k_{max}$, set to the corresponding value. Such a strategy could be useful when accuracy is very important.

There are several directions in which this research can continue. One direction is to combine the previous and current versions of the accuracy-enhancing solver. Specifically, after two-three iterations with ArLyap, one can switch to the computations updating the solution of the original equation at each of the next iterations. Another direction is to refine the stopping strategy, by allowing the iterative process to continue if the normalized residual at a certain iteration exceeds its value at the previous iteration by more than, e.g., two times, but stop the process by restoring the previous iterate at the second detection of a residual increase. Finally, it could be tried to perform the back transformation in quadruple precision. The IEEE standard 754-2008 specifies quadruple and even octuple precision, and some Fortran compilers allow quadruple precision. Moreover, it could be worth trying to make full computations in Fortran, including data input and evaluation of the results. It is expected that better accuracy will be obtained this way.

## 4. Materials and Methods

The fact that Lyapunov equations retain only the linear part of AREs suggested that some ARE solvers might be specialized for solving them. Previous successful experience with the algorithms for AREs based on Newton's method, with or without line search [36–38], recommended them as good candidates. Recently, the author adapted the Newton-based ARE solver to Lyapunov equations. The conceptual algorithm in [35] is briefly discussed in following subsection and further improved in the next subsections for achieving the highest efficiency.

### 4.1. Conceptual Algorithm Description

Starting from a given initial solution, $X_0$, or with $X_0 = 0$, the algorithm computes the current residual matrix (at iteration $k$), $\mathcal{R}(X_k)$, defined as

$$\mathcal{R}(X_k) := \text{op}(A)^T X_k \, \text{op}(E) + \text{op}(E)^T X_k \, \text{op}(A) + Y, \tag{10}$$
$$\mathcal{R}(X_k) := \text{op}(A)^T X_k \, \text{op}(A) - \text{op}(E)^T X_k \, \text{op}(E) + Y, \tag{11}$$

for a continuous- or discrete-time equation, respectively. Then, a generalized (or standard, if $E = I_n$) Lyapunov Equation (12) or (13), respectively, which has the current residual matrix in the right hand side,

$$\text{op}(A)^T L_k \, \text{op}(E) + \text{op}(E)^T L_k \, \text{op}(A) = -\mathcal{R}(X_k), \tag{12}$$

$$\text{op}(A)^T L_k \, \text{op}(A) - \text{op}(E)^T L_k \, \text{op}(E) = -\mathcal{R}(X_k), \tag{13}$$

is solved in $L_k$, and the current solution is updated, $X_{k+1} = X_k + L_k$.

The main termination criterion for the iterative process is defined based on the *normalized residual*, $r_k := r(X_k)$, and a tolerance $\tau$. Specifically, if

$$r_k := \|\mathcal{R}(X_k)\|_F / \max(1, \|X_k\|_F) \le \tau, \tag{14}$$

the computations are terminated with the computed solution $X_k$. A default tolerance is used if $\tau \le 0$ is given on input. Its value is defined by one of the formulas below for Equations (4) and (5), respectively,

$$\begin{aligned}
\tau &= \min\left\{ \varepsilon_M \sqrt{n} \left( 2 \|A\|_F \|E\|_F + \|Y\|_F \right), \sqrt{\varepsilon_M}/10^3 \right\}, \\
\tau &= \min\left\{ \varepsilon_M \sqrt{n} \left( \|A\|_F^2 + \|E\|_F^2 + \|Y\|_F \right), \sqrt{\varepsilon_M}/10^3 \right\}.
\end{aligned} \tag{15}$$

Another termination criterion is the MATLAB-style *relative residual*, $r_r(X_k)$, defined as the ratio between $\|\mathcal{R}(X_k)\|_F$ and the sum of the Frobenius norms of the matrix terms in Equation (4) or (5). In addition, if $\|L_k\|_F \le \varepsilon_M \|X_k\|_F$ the iterative process terminates with the computed solution $X_k$.

For increased efficiency, $A$ and $E$ are reduced at iteration $k = 0$ to the real Schur-triangular form, using two orthogonal transformations, $Q$ and $Z$, namely

$$\widetilde{A} = Q^T A Z, \quad \widetilde{E} = Q^T E Z, \tag{16}$$

where $\widetilde{A}$ is block upper triangular with diagonal blocks or order 1 and 2, corresponding to real and complex conjugate eigenvalues, respectively, and $\widetilde{E}$ is upper triangular. Then, the right hand side of Equation (12) or (13) is transformed

$$\widetilde{\mathcal{R}}(X_k) := Z^T \mathcal{R}(X_k) Z, \text{ if } \text{op}(M) = M, \quad \text{or} \quad \widetilde{\mathcal{R}}(X_k) := Q^T \mathcal{R}(X_k) Q, \text{ if } \text{op}(M) = M^T. \tag{17}$$

A so-called *reduced equation*, Equation (18) or (19),

$$\text{op}(\widetilde{A})^T \widetilde{L}_k \, \text{op}(\widetilde{E}) + \text{op}(\widetilde{E})^T \widetilde{L}_k \, \text{op}(\widetilde{A}) = -\widetilde{\mathcal{R}}(X_k), \tag{18}$$

$$\text{op}(\widetilde{A})^T \widetilde{L}_k \, \text{op}(\widetilde{A}) - \text{op}(\widetilde{E})^T \widetilde{L}_k \, \text{op}(\widetilde{E}) = -\widetilde{\mathcal{R}}(X_k), \tag{19}$$

respectively, is solved for $\widetilde{L}_k$. Finally, $\widetilde{L}_k$ is back transformed,

$$L_k = Q \widetilde{L}_k Q^T, \text{ if } \text{op}(M) = M, \quad \text{or} \quad L_k = Z \widetilde{L}_k Z^T, \text{ if } \text{op}(M) = M^T, \tag{20}$$

and used to improve the current solution estimate, $X_k$.

### 4.2. New Algorithm

It will now be shown that it is not necessary to transform the solution of the reduced Lyapunov equations, $\widetilde{L}_k$, back to $L_k$, except for the final iteration. Indeed, using the notation introduced above, let $\widetilde{X}_k := Q^T X_k Q$, if $\text{op}(M) = M$, and $\widetilde{X}_k := Z^T X_k Z$, if $\text{op}(M) = M^T$. For brevity, only the first case will be considered, since the second case is similar. From Equation (16), it follows that $A = Q \widetilde{A} Z^T$, and $E = Q \widetilde{E} Z^T$, so that replacing $A$ and $E$ in Equation (4), we get

$$Z \widetilde{A}^T Q^T X Q \widetilde{E} Z^T + Z \widetilde{E}^T Q^T X Q \widetilde{A} Z^T = -Y,$$

and premultiplying by $Z^T$, postmultiplying by $Z$, and setting $\widetilde{X} := Q^T X Q$, $\widetilde{Y} := Z^T Y Z$, this formula becomes

$$\widetilde{A}^T \widetilde{X} \widetilde{E} + \widetilde{E}^T \widetilde{X} \widetilde{A} = -\widetilde{Y}. \tag{21}$$

Similarly, Equations (10) and (17) imply

$$\widetilde{\mathcal{R}}(X_k) := Z^T \mathcal{R}(X_k) Z = \widetilde{A}^T \widetilde{X}_k \widetilde{E} + \widetilde{E}^T \widetilde{X}_k \widetilde{A} + \widetilde{Y}. \tag{22}$$

But from Equation (18),

$$-\widetilde{\mathcal{R}}(X_k) = \widetilde{A}^T \widetilde{L}_k \widetilde{E} + \widetilde{E}^T \widetilde{L}_k \widetilde{A}.$$

Adding the last two equations, it follows that $\widetilde{X}_{k+1} := \widetilde{X}_k + \widetilde{L}_k$ solves Equation (21), hence $X_{k+1} = X_k + L_k$ theoretically solves Equation (4). Since $X_k$ and $\widetilde{X}_k$ are related by a similarity transformation ($\widetilde{X}_k := Q^T X_k Q$ or $\widetilde{X}_k := Z^T X_k Z$), which preserves their eigenvalues, it follows that $\|X_k\|_F = \|\widetilde{X}_k\|_F$. The same is true for $\mathcal{R}(X_k)$ and $\widetilde{\mathcal{R}}(X_k)$. Therefore, the normalized residuals for $X_k$ and $\widetilde{X}_k$ also coincide (from Equation (9) with $\widehat{X}$ and $X_m$ replaced by $X_k$). The same argument shows that the tolerance $\tau$ in Equation (15), computed for the given matrices, $A$, $E$, and $Y$, coincides with its value computed for the transformed matrices, $\widetilde{A}$, $\widetilde{E}$, and $\widetilde{Y}$. This proves that the whole iterative process can be performed solving only reduced Lyapunov equations. Just at the final iteration, after convergence, the solution of the reduced equation should be used for computing $Q(\widetilde{X}_k + \widetilde{L}_k)Q^T$.

The same arguments as above can be used for solving Equation (4) with $\text{op}(A) = A^T$, or for solving discrete-time Lyapunov Equation (5).

It is important to emphasize that, in theory, there is no need for an iterative process, but this can be useful in practice, due to numerical errors and possibly bad numerical conditioning of a Lyapunov equation.

The new algorithm can be stated as Algorithm 1.

---

**Algorithm 1** Algorithm ArLyap: Accuracy-enhancing Lyapunov solver

---

**Input:** The matrices $A$, $E$, and $Y$, and an integer $k_{\max}$; optionally, initial $X_0$ and a tolerance $\tau$.
**Ensure:** The solution $X_k$ of Equations (4) or (5).
1: Compute $\widetilde{A} = Q^T A Z$, $\widetilde{E} = Q^T E Z$, and $\widetilde{Y} = Z^T Y Z$, if $\text{op}(M) = M$, or $\widetilde{Y} = Q^T Y Q$, if $\text{op}(M) = M^T$.
2: If $X_0$ is given, evaluate $\widetilde{X}_0 = Q^T X_0 Q$, if $\text{op}(M) = M$, or $\widetilde{X}_0 = Z^T X_0 Z$, if $\text{op}(M) = M^T$.
3: Otherwise, set $\widetilde{X}_0 = 0$.
4: **for** $k = 0, 1, \ldots, k_{\max}$ **do**
5:     Compute the residual matrix $\widetilde{\mathcal{R}}(X_k)$

$$\widetilde{\mathcal{R}}(X_k) := \text{op}(\widetilde{A})^T \widetilde{X}_k \, \text{op}(\widetilde{E}) + \text{op}(\widetilde{E})^T \widetilde{X}_k \, \text{op}(\widetilde{A}) + \widetilde{Y},$$
$$\widetilde{\mathcal{R}}(X_k) := \text{op}(\widetilde{A})^T \widetilde{X}_k \, \text{op}(\widetilde{A}) - \text{op}(\widetilde{E})^T \widetilde{X}_k \, \text{op}(\widetilde{E}) + \widetilde{Y},$$

6:     for Equation (4) or (5), respectively.
7:     If $r_k := \|\widetilde{\mathcal{R}}(X_k)\|_F / \max(1, \|\widetilde{X}_k\|_F) \leq \tau$, exit the loop.
8:     Solve in $\widetilde{L}_k$ the reduced Lyapunov Equation (18) or (19), respectively.
9:     Update $\widetilde{X}_{k+1} = \widetilde{X}_k + \widetilde{L}_k$.
10: **end for**
11: Compute $X_k = Q \widetilde{X}_k Q^T$, if $\text{op}(M) = M$, or $X_k = Z \widetilde{X}_k Z^T$, if $\text{op}(M) = M^T$ and return $X_k$.
12: If $k = k_{\max}$, "Convergence has not been achieved."

---

*4.3. Computational Modules for Improving Efficiency*

Solving only reduced Lyapunov equations decreases the computational effort with about $1.5n^3$ floating point operations (flops) per iteration, by avoiding the back transformation of $\widetilde{L}_k$ to $L_k$ in Equation (20). (This evaluation assumes that the symmetry is exploited.) Additional gains in

efficiency can be obtained by simplifying the computation of residuals, since $\widetilde{A}$ is in a Schur form, and $\widetilde{E}$ is upper triangular. Before commenting on how these improvements could be obtained, few remarks come in order. It is worth mentioning that high-quality numerical software makes references only to the needed part of an array storing a matrix. For instance, only the entries of an upper (or lower) triangle of a symmetric matrix are referenced. All elements on the first subdiagonal of a real Schur matrix are also referenced, and the position of its zero values defines the $1 \times 1$ or $2 \times 2$ blocks (needed, e.g., for computing the eigenvalues). Note that a matrix in upper Schur form is a special case of an upper Hessenberg matrix which has no two consecutive nonzero subdiagonal elements.

A professional implementation of the ArLyap solver would need to consider several basic computational modules, which are not available in BLAS [17], LAPACK [18], or SLICOT libraries. Specifically, such modules are described below.

1.  Compute $R := \alpha R + \beta(\operatorname{op}(H)^T X + X \operatorname{op}(H))$, with $H$ an upper Hessenberg matrix and $X$ a symmetric matrix. This is a special symmetric "rank 2k operation" (a specialized version of the BLAS 3 routine syr2k), needed, e.g., for solving standard continuous-time reduced Lyapunov Equation (18), with $\widetilde{E} = I_n$.

2.  Compute $R := \alpha R + \beta \operatorname{op}(H)^T X \operatorname{op}(H)$, with $H$ an upper Hessenberg matrix and $X$ a symmetric matrix. This operation is necessary for solving standard or generalized discrete-time reduced Lyapunov Equation (19). Let $\operatorname{diag}(X)$, $\operatorname{triu}(X)$, and $\operatorname{tril}(X)$ denote the diagonal, upper and lower triangles of $X$, respectively, and define two, upper and lower, respectively, triangular matrices

    $$U := \operatorname{triu}(X) - \operatorname{diag}(X)/2, \quad \text{if } \operatorname{op}(H) = H,$$
    $$L := \operatorname{tril}(X) - \operatorname{diag}(X)/2, \quad \text{if } \operatorname{op}(H) = H^T.$$

    Since

    $$X = U + L, \quad U = L^T, \tag{23}$$

    it follows that

    $$\left.\begin{array}{l} H^T X H = H^T(UH) + (UH)^T H, \\ HXH^T = (HU)H^T + H(HU)^T, \end{array}\right\} \quad \text{if upper triangular part of } X \text{ is used,}$$

    $$\left.\begin{array}{l} H^T X H = H^T(L^T H) + (L^T H)^T H, \\ HXH^T = (HL^T)H^T + H(HL^T)^T, \end{array}\right\} \quad \text{if lower triangular part of } X \text{ is used.}$$

    Above, $UH$, $HU$, $L^T H$, and $HL^T$ are again upper Hessenberg matrices. Therefore, all four formulas above are special cases of symmetric rank 2k operations, where both matrices involved are in upper Hessenberg form. Note that one could define, e.g.,

    $$H^T X H = (LH)^T H + H^T(LH), \quad \text{if lower triangular part of } X \text{ is used,}$$

    but then the matrix $LH$ would be a full matrix, hence more computational effort would be needed to evaluate $H^T X H$.

3.  Compute $R := \alpha R + \beta(\operatorname{op}(H) \operatorname{op}(G)^T + \operatorname{op}(G) \operatorname{op}(H)^T)$, with $H$ and $G$ upper Hessenberg matrices. This module is called by the module 2.

4. Compute $R := \alpha R + \beta \operatorname{op}(E)^T X \operatorname{op}(E)$, with $E$ an upper triangular matrix and $X$ a symmetric matrix. This operation is needed for solving generalized discrete-time reduced Lyapunov Equation (19), and it can be performed using the formulas:

$$\left.\begin{array}{l} E^T X E = E^T(UE) + (UE)^T E, \\ EXE^T = (EU)E^T + E(EU)^T, \end{array}\right\} \quad \text{if } U \text{ is used,}$$

$$\left.\begin{array}{l} E^T X E = E^T(L^T E) + (L^T E)^T E, \\ EXE^T = (EL^T)E^T + E(EL^T)^T, \end{array}\right\} \quad \text{if } L \text{ is used.}$$

   Note that $UE$, $EU$, $L^T E$, and $EL^T$ are all upper triangular matrices. Hence, each of these four formulas involve a special symmetric rank 2k operation on an upper triangular pair. This module needs the product of two upper triangular matrices, expressed as $UE$, or $EU$, or $L^T E$, or $EL^T$, with $U$ and $E$ upper triangular, and $L$ lower triangular. This is easily done internally using BLAS 2 function trmv in a loop with $n$ cycles.

5. Compute $R := \alpha R + \beta(\operatorname{op}(E)\operatorname{op}(U)^T + \operatorname{op}(U)\operatorname{op}(E)^T)$, with $E$ and $U$ upper triangular matrices. This module is called by the module 4.

6. Compute

$$\begin{array}{ll} P = HX, & \text{if } \operatorname{op}(M) = M, \\ P = XH, & \text{if } \operatorname{op}(M) = M^T, \end{array} \tag{24}$$

   with $H$ an upper Hessenberg matrix and $X$ a symmetric matrix, given either the upper triangle $U$ or the lower triangle $L$ of $X$. This module is needed for computing the relative residual for standard continuous-time reduced Lyapunov equations, since it allows to evaluate the Frobenius norm of this matrix product (which is a term of that equation). Using $X = U + \tilde{U}^T$, or $X = L^T + \tilde{L}$, where $\tilde{U}$ and $\tilde{L} = \tilde{U}^T$ are strictly upper and lower triangular, respectively, the module evaluates the product using BLAS 2 trmv function and other routines. Clearly, both $HU$ and $L^T H$ are upper Hessenberg, but the results of this module are full matrices. Using Equation (24), the function of the module 1 becomes $R := \alpha R + \beta(P + P^T)$. However, this formula should only be used when relative residual is needed, and hence $P$ should be computed.

7. Compute

$$R := \alpha R + \beta(\operatorname{op}(H)^T X \operatorname{op}(E) + \operatorname{op}(E)^T X \operatorname{op}(H)), \tag{25}$$

   with $H$ an upper Hessenberg matrix, $X$ a symmetric matrix, and $E$ an upper triangular matrix. This operation is needed for solving generalized continuous-time reduced Lyapunov Equation (18). Using Equation (23), it follows that

$$\left.\begin{array}{l} H^T X E + E^T X H = H^T(UE) + (UE)^T H + (UH)^T E + E^T(UH), \\ HXE^T + EXH^T = (HU)E^T + E(HU)^T + H(EU)^T + (EU)H^T, \end{array}\right\} \quad \text{if } U \text{ is used,}$$

$$\left.\begin{array}{l} H^T X E + E^T X H = H^T(L^T E) + (L^T E)^T H + (L^T H)^T E + E^T(L^T H), \\ HXE^T + EXH^T = (HL^T)E^T + E(HL^T)^T + H(EL^T)^T + (EL^T)H^T, \end{array}\right\} \quad \text{if } L \text{ is used.}$$

   where $X = U + L$, and $U = L^T$. Note that $UE$, $EU$, $L^T E$, and $EL^T$ are all upper triangular, and $UH$, $HU$, $L^T H$, and $HL^T$ are all upper Hessenberg. Consequently, each of these four formulas involve two special symmetric rank 2k operations for upper Hessenberg-triangular pairs.

8. Compute $R := \alpha R + \beta(\operatorname{op}(H)^T \operatorname{op}(E) + \operatorname{op}(E)^T \operatorname{op}(H))$, with $H$ an upper Hessenberg matrix and $E$ an upper triangular matrix. This operation is called by the module 7.

9. Compute either $P$ or $P^T$, where $P := \operatorname{op}(H)^T X \operatorname{op}(E)$, with $H$ an upper Hessenberg matrix, $X$ a symmetric matrix, and $E$ an upper triangular matrix. This module is needed for evaluating the Frobenius norm of $P$, used to obtain the relative residual for generalized continuous-time reduced

Lyapunov equations. The matrix $R$ in Equation (25) becomes $R := \alpha R + \beta(P + P^T)$. However, this formula should only be used when relative residual is needed. Note that $P$ is a general matrix, with no structure. The computations can be performed as follows: using the module 6, compute $W = HX$, if $\text{op}(M) = M$, or $W = XH$, if $\text{op}(M) = M^T$; then, compute $P = WE^T$, if $\text{op}(M) = M$, or $P = E^T W$, if $\text{op}(M) = M^T$, using a BLAS 3 trmm operation. Note that the Frobenius norms of $P$ and $P^T$ coincide, and $R$ can be obtained having either $P$ or $P^T$.

All modules operating with a symmetric matrix must use either the upper, or the lower triangle of an array storing that matrix. Similarly, for an upper Hessenberg matrix, the entries below the first subdiagonal should not be referenced. The modules discussed above represent an extension of the BLAS library, extension which is important for the ArLyap solver, but can be used for other applications as well.

For large order Lyapunov equations, it would be necessary to provide block variants for some of the modules above. As an example, consider the operation $HX$, with $H$ upper Hessenberg, and $X$ symmetric. Since in the ArLyap solver, $H$ is actually in a real Schur form, let us partition,

$$HX = \begin{bmatrix} H_{11} & H_{12} \\ 0 & H_{22} \end{bmatrix} \begin{bmatrix} X_{11} & X_{12} \\ X_{12}^T & X_{22} \end{bmatrix} = \begin{bmatrix} H_{11}X_{11} + H_{12}X_{12}^T & H_{11}X_{12} + H_{12}X_{22} \\ H_{22}X_{12}^T & H_{22}X_{22} \end{bmatrix}, \qquad (26)$$

where $H_{ii} \in \mathbb{R}^{n_i \times n_i}$, $i = 1, 2$, $n_1 + n_2 = n$, and $H_{n_1+1,n_1} = 0$. Clearly, $H_{11}X_{11}$ and $H_{22}X_{22}$ can be computed with the module 6, $H_{12}X_{12}^T$ requires a BLAS 3 operation gemm, $H_{11}X_{12}$ and $H_{22}X_{12}^T$ can be evaluated with an easy extension of the BLAS 3 operation trmm, and $H_{12}X_{22}$ is obtained by BLAS 3 operation symm. These ideas can be generalized for finer partitions and for other modules above.

## 5. Conclusions

A new accuracy-enhancing solver for standard and generalized continuous- and discrete-time Lyapunov equations, has been proposed and investigated. The underlying algorithm and some technical details have been summarized. The best available algorithms for solving Lyapunov equations with dense coefficient matrices, based on the orthogonal reduction to the real Schur(-triangular) form are used in the implementation. The Schur(-triangular) reduction is performed only once, before starting the iterative process. During the iterations, reduced Lyapunov equations are solved. The result of the last iteration is back transformed to obtain the solution of the original equation. How the computations can be organized to increase the efficiency by exploiting the structure and symmetry is also detailed. The numerical results found when solving series of numerically difficult examples generated using SLICOT benchmark collections CTLEX and DTLEX are compared to the solutions computed by the MATLAB and SLICOT solvers. The ArLyap solver can be more accurate than the other solvers, especially for ill-conditioned equations, without a significant additional computational effort. Actually, with very few exceptions, the ArLyap solver is faster than the MATLAB solvers, and close to the SLICOT solvers regarding the elapsed CPU times.

**Funding:** This research was partially funded by the Ministry of Research and Innovation, Romania, Institutional research programme PN 1819, project PN 1819-01-01.

**Conflicts of Interest:** The author declares no conflict of interest. The funders had no role in the design of the study; in the collection, analyses, or interpretation of data; in the writing of the manuscript, or in the decision to publish the results.

## Abbreviations

The following abbreviations are used in this manuscript:

| | |
|---|---|
| ARE | algebraic Riccati equation |
| CTLEX | continuous-time Lyapunov equation |
| DTLEX | discrete-time Lyapunov equation |
| TLEX | continuous or discrete-time Lyapunov equation |

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
