# Peer review of "Comparative Performance Evaluation of an Accuracy-Enhancing Lyapunov Solverâ€"

_information, doi:10.3390/info10060215_

Round 1
Reviewer 1 Report
The paper brings nice contributions to the CACSD field, by constructing a deeper insight into the resolution of continuous- and discrete-time Lyapunov equations, in standard and generalized forms. The key elements of the new approach refer to the highly-performant exploitation of the Newton-based solvers. The developments rely on the author’s experience in the area, already reported in some previous publications investigating the Riccati equations, such as [25]-[27]. The benefits of the proposed algorithm are discussed in terms of relevant comparisons with Lyapunov solvers available in well-known environments for scientific computations, by using recognized collections of test problems. The paper reads well, in spite of the very specialized level of the considered problems; the elegant exposition style is able to create multiple facilities for bridging common to advanced topics.
Author Response
I thank the reviewer for his positive comments.
Reviewer 2 Report
See the attachment.

Author Response
Many thanks for the constructive suggestions and comments!
I will describe the changes I have made.
Issue 1. "There exist many numerical algorithms devoted to solving the standard and generalized Lyapunov equations (see, e.g., [SZ03], [Kaa11], [Sav14] and references therein). The author should update the reference list and should compare its algorithm with the latest such algorithms or at least with some of the classical ones as Bartels-Stewart method, Hammarling method, Hessenberg-Schur method etc."
I added the three references listed above and other references (e.g., Golub, Nash, and Van Loan (1979), Jonsson and Kagstrom (2002), Simoncini (2016), etc.), so that the reference list increased from 32 to 41 references. I added a text in Section 1 (lines 63-94), which briefly describes other approaches, including iterative algorithms exploiting sparsity and/or the low-rank structure. All solvers investigated in the paper (including MATLAB lyap and dlyap functions and SLICOT routines) are of the Bartels-Stewart type, and the accuracy-enhancing solvers (previous solver, ALyap and the new one, ArLyap) use them at each iteration. Hammarling's algorithm can relatively easily be called instead in ArLyap. However, it is less general. These issues are mentioned in the lines 101-104 of the revised manuscript. The Hessenberg-Schur algorithm, mentioned now in the lines 71-74, offers no advantage over Bartels-Stewart algorithm for Lyapunov equations.
Issue 2. "Although the author claims that the presented algorithm is an improvement of previous one reported in his own paper [24], the exhibited data for Ex. 4.1 and especially for Ex. 4.3 show exactly the opposite. Indeed, in many cases the present method gives worse results for the relative errors, CPU times and iteration numbers. The author should discuss on this and stress on the advantages and disadvantages of the new method."
I refined the text so that the "improvement" only refer to the computational efficiency, improvement obtained by exploiting the structure of the reduced Lyapunov equations. For most examples, the normalized residuals for these equations are smaller than those corresponding to the original equations, and the number of iterations needed for achieving the convergence is almost always smaller than for the previous solver, ALyap. These issues are highlighted in the lines 201-210 for CTLEX 4.1. It is noticed that for ill-conditioned examples, the new solver, ArLyap, can obtain smaller relative errors than ALyap. Moreover, Table 1 (after line 210) illustrates that less iterations may be needed by ArLyap. For CTLEX 4.3 examples, ArLyap is comparable with the SLICOT solver, and less accurate than ALyap, but in many cases it is much more accurate than MATLAB lyap. The reason of possible lower accuracy compared to ALyap is explained in Section 3. A substantial part has been added in the revised version (lines 286-344). This explanation supports the experimental findings. Several directions for continuing the research have also been mentioned (lines 345-355).
Issue 3. "It is often hard to follow the manuscript since the reader is forced to look for definitions, formulas and comments forward in this article or in other publications. So that, the author should rearrange the paper to rid the inconveniences. I suggest the definitions of often used notions as op(M), R(M), `Frobenius norm', `specific example index' etc. to be given somewhere in the beginning of the paper. Also, in the whole work the author should use only one name for the investigated method, otherwise he confuses the reader."
I added a text in the lines 120-137, defining op(M), R(M), `Frobenius norm', and other often used notions and notation. "Specific example index" is no longer used as such, but a reformulation appears in the lines 169-170. I used "ArLyap" for all references to the new algorithm and solver, and ALyap for the previous version. All occurrences of "Newton" in the previous figure legends have been replaced by ArLyap.
Issue 4. "If there is no mistake in the formula (23), then the author should explain H_{12} X_{12}^T = 0."
This was indeed a mistake, which has been corrected out. Thanks for pointing it out!
Issue 5. "Finally, more detailed discussion on the exhibited graphs is needed."
I added the lines 201-220 for CTLEX 4.1 series of examples, including Table 1, and a discussion about the "privileged" position of MATLAB solvers in comparison with the other solvers (lines 211-220).
I added a phrase in the lines 223-224 for CTLEX 4.2, a new text in the lines 235-239 for DTLEX 4.1, a phrase in the lines 242-243 for DTLEX 4.2, a phrase in the lines 246-247 for DTLEX 4.3, and the text in the lines 253-256 for DTLEX 4.4.
All additional comments have been considered.
Comment 1: "The author could emphasize on timeliness and relevance of the study by spending couple of more words about the iterative methods and their rapid development in 21st century. References to some recent studies on the most famous among these methods are reasonable. For example, [KYI17], [AMOS19] and some of the references therein can be used."
The suggested references have been added to the reference list and a related text has been added in the lines 89-94.
Comment 2. "The last sentence in Line 80 should be dropped since it is a necessary
condition for a scientific paper."
The sentence has been deleted (in the line 115).
Comment 3. "It would be better if the modules in Subsection 4.3 were numbered."
The modules have been numbered, and they are referenced by those numbers.
Comment 4. "The formulas in Line 282 should be numbered and quoted."
The formulas have been numbered as (23), before the line 441, and quoted in the line 472.
Comment 5. "The sentence `Denoting . . . ', in lines 306-307, should be dropped and the mentioned formula should be numbered and quoted."
The sentence `Denoting . . . ', in the former lines 306-307 has been deleted and the formula
has been numbered as (24), after the line 459, and referred to in the line 466.
Comment 6. "It would be better the sentence `This module . . . ', in lines 319-321, to be moved at the end of the paragraph."
The sentence has been reformulated in the lines 479-481, but it has not been moved at the end of the paragraph because this is the essential text about the module 9.
Comment 7. "The sentence `The function . . . ', in lines 321-322, should be dropped
and the mentioned formula should be quoted."
The sentence has been replaced as it appears now in line 481, referring to the formula (25).
Reviewer 3 Report
The work seems interesting and meaningful. I have only one question: Can the method be used for constructing Lyapunov functions for nonlinear systems? If not, could you please provide some future works or possible tendency? This would be interesting.
Author Response
I thank the reviewer for his comments.
I don't yet know if the method could be used for constructing Lyapunov functions for nonlinear systems.
There are some references about this topic, e.g.,
1. Carla A Schwartz, Aiguo Yan. Construction of Lyapunov Functions for Nonlinear Systems Using Normal Forms. Journal of Mathematical Analysis and Applications, Volume 216, Issue 2, 15 December 1997, Pages 521-535, where a systematic method for this construction is given.
2. Peter Giesl, Sigurdur Hafstein. Construction of Lyapunov functions for nonlinear planar systems by linear programming, Journal of Mathematical Analysis and Applications, Volume 388, Issue 1, 1 April 2012, Pages 463-479.
I did not haave the opportunity to investigate this issue.
Round 2
Reviewer 2 Report
I am satisfied with the corrections made and I think the paper could be considered for publication in Information.
Before publishing only one more correction is needed.
Namely, the list of often used notions (rows 128-145) should be moved before formula (4).

Author Response
To follow your suggestion, I moved the list of often used notions before formula (4).
Many thanks again for all your suggestions and comments, which contributed to a significant improvement of the manuscript!